# TOSS: HIGH-QUALITY TEXT-GUIDED NOVEL VIEW SYNTHESIS FROM A SINGLE IMAGE

**Yukai Shi**[1,3*†]   **Jianan Wang**[3*]   **He Cao**[2,3*†]

**Boshi Tang**[1,3]   **Xianbiao Qi**[3]   **Tianyu Yang**[3]   **Yukun Huang**[3]   **Shilong Liu**[1,3]

**Lei Zhang**[3]   **Heung-Yeung Shum**[1,3]

[1] Tsinghua University   [2] Hong Kong University of Science and Technology
[3] International Digital Economy Academy (IDEA)

(a) TOSS increases plausibility with text

(b) TOSS increases controllability with text

(c) TOSS generates novel views with higher quality and multiview-consistency

(d) TOSS generates novel views with higher quality (random views)

Zero123   TOSS   GT

Figure 1: **TOSS** significantly boosts novel view *plausibility* with additional textual guidance (a) and grants users a better *controllability* over concealed parts of an object (b). We demonstrate higher novel view generation quality and *multiview-consistency* with look-around views (c) and random views (d).

## ABSTRACT

In this paper, we present TOSS, which introduces text to the task of novel view synthesis (NVS) from just a single RGB image. While Zero123 has demonstrated impressive zero-shot open-set NVS capability, it treats NVS as a pure image-to-image translation problem. This approach suffers from the challengingly under-constrained nature of single-view NVS: the process lacks means of explicit user control and often results in implausible NVS generations. To address this limitation, TOSS uses text as high-level semantic information to constrain the NVS solution space. TOSS fine-tunes text-to-image Stable Diffusion pre-trained on large-scale text-image pairs and introduces modules specifically tailored to image and camera pose conditioning, as well as dedicated training for pose correctness and preservation of fine details. Comprehensive experiments are conducted with results showing that our proposed TOSS outperforms Zero123 with more plausible, controllable and multiview-consistent NVS results. We further support these results with comprehensive ablations that underscore the effectiveness and potential of the introduced semantic guidance and architecture design.

---

*Equal contribution.

†Work done during an internship at IDEA.

# 1 INTRODUCTION

Novel view synthesis (NVS) from a single RGB image is a severely under-constrained problem, which requires imagining an object's unobserved geometry and textures. Despite of its ill-posed nature, single-view NVS aims to change the viewpoint of an object given just a single image. Therefore it has been regarded as one of the foundation tasks and prerequisite capability for 3D generation (Qian et al., 2023; Liu et al., 2023a), especially when generalizing to any object captured by users or generated by text-to-image models (Rombach et al., 2022).

Recent advances in modeling, especially score-based diffusion models (Ho et al., 2020; Sohl-Dickstein et al., 2015) have sparked interest in approaching novel view synthesis (NVS) as an image-to-image translation problem (Chan et al., 2023; Watson et al., 2022). Pioneering work of Zero123 (Liu et al., 2023b) extends the generalization capability from a single category to the open world. In concrete, it leverages an image-to-image model fine-tuned from Stable Diffusion (Rombach et al., 2022) and learns to control relative camera transformation during the generation process by further fine-tuning on image renderings derived from a 3D dataset (Deitke et al., 2023). It is worth noting that Stable Diffusion is pre-trained on large-scale web data (Schuhmann et al., 2022), which empowers Zero123 (Liu et al., 2023b) with impressive zero-shot open-set NVS capability.

However, current NVS methods often exhibit shape and texture deformations that lead to implausible semantic meaning (Liu et al., 2023c) of their results: *e.g.* the generated novel view of a shark may have two tails as demonstrated in Fig. 1 (a). The reason for the poor results is rooted in the ill-posed nature of single-view NVS, where there exist multiple solutions for the unobserved geometry and textures. Previous works constrain the solution space with the input view image and the camera pose. However, the NVS solution space constrained by the two conditions is still too large to guarantee a plausible NVS result. Moreover, even when the generation is plausible, users still desire more means to control the NVS process, otherwise the generated result may be visually plausible but fail to fulfill their expectation as shown in Fig. 1 (b).

To address the issue of implausible semantic meaning and to add more controllability to NVS, we introduce ***TOSS***, a high-quality text-guided NVS model that explicitly leverages textual description as semantic guidance. Before we introduce more algorithm design, it is worth pointing out the benefits of TOSS: 1) Text can impose constraints on the unobservable parts of an object, granting users explicit control to narrow down the NVS solution space as desired; 2) Text can serve as high-level semantic information that complements the low-level details provided by images, which can act as a strong prior for generating plausible NVS results; 3) Users can further refine the focus and granularity of the text description, providing an explicit and flexible means to constrain the NVS solution space according to generation requirements. By introducing text information, we observe a significantly improved quality of NVS results. This improvement is attributed to a more informative exploration of the optimization landscape with extra semantic guidance.

More specifically, as we aim to synthesize novel view images under the guidance of text semantics, naturally we capitalize on Stable Diffusion (Rombach et al., 2022) for its remarkable ability to generate diverse images based on text descriptions. To adapt Stable Diffusion for the NVS task, we modify it by introducing a novel dense cross attention module to condition the generative process on both the input image's feature maps produced by Stable Diffusion's VAE encoder, and the frequency encoding on the relative camera pose transformation. We fine-tune our proposed model TOSS on the 3D dataset Objaverse (Deitke et al., 2023). To achieve higher-quality NVS results while maintaining the same training and inference computation cost, we use a training strategy that efficiently specializes two expert diffusion models to focus on novel view pose correctness and preservation of details. Moreover, our improvements are orthogonal to and can be readily combined with advancements for text-to-image generation, *e.g.* better diffusion models, textual inversion (Gal et al., 2022), *etc.*, to further boost the performance.

The main contributions of this paper can be summarized as:

- We introduce TOSS, a zero-shot open-set novel view synthesis (NVS) diffusion model that explicitly uses text to narrow down the NVS solution space to yield more plausible, controllable and multiview-consistent results, given just a single RGB image.

- We propose image and camera pose conditioning modules that are meticulously tailored to the text-guided NVS model, alongside training strategies that dedicate specialized denoising experts for novel view pose correctness and preservation of fine details.

- Extensive experiments demonstrate that TOSS exhibits superiority to Zero123 on the tasks of NVS and 3D reconstruction, in terms of plausibility, controllability and multiview-consistency. Furthermore, we empirically showcase the substantial potential inherent to our approach, including improved quality through the utilization of higher-capability text-to-image models and techniques such as textual inversion.

## 2 RELATED WORK

**2D Diffusion Model.** Diffusion models (Ho et al., 2020; Croitoru et al., 2023) have demonstrated remarkable performance in generating 2D images. Their stable training objectives and enhanced distribution mode coverage have enabled diffusion models to surpass previous generative models (Goodfellow et al., 2020). Numerous studies have explored modeling conditional distributions (Song et al., 2021; Dhariwal & Nichol, 2021) with diffusion models, such as text-to-image generation (Ramesh et al., 2022; Nichol et al., 2021) and image super-resolution (Saharia et al., 2022b; Ho et al., 2022). Large effort has been devoted to training text-to-image models on large-scale web datasets (Changpinyo et al., 2021; Schuhmann et al., 2021; 2022), equipping such pre-trained models (Rombach et al., 2022; Ramesh et al., 2022; Saharia et al., 2022a) with rich real-world knowledge. Subsequent works (Li et al., 2023b; Zhang et al., 2023) consequently leverage powerful pre-trained diffusion models for controllable generation, *e.g.* layout-to-image, pose-to-image and etc.

**Novel View Synthesis from a Single Image.** Novel view synthesis from a single RGB image requires imagining an object's unobserved geometry and texture, which is a precursor to 3D generation despite its ill-posed nature. *Regression-based* methods enhance NVS quality with geometry priors, but often with blurry novel view results because of the mean-seeking nature (Yu et al., 2021; Le et al., 2020; Or-El et al., 2022; Park et al., 2019). *Generation-based* methods leverage generative capabilities to produce high-fidelity novel view predictions. GAN-based approaches (Chan et al., 2021; Noguchi et al., 2022) can generate high-resolution novel view images, but suffer from unstable training. Liu et al. (2023b); Watson et al. (2022); Chan et al. (2023) leverage diffusion models' stronger modelling capability and training stability for higher-quality NVS results. Nevertheless, their efficacy remains confined to specific categories. Recently, Zero123 (Liu et al., 2023b) pioneers in zero-shot open-set NVS utilizing pre-trained diffusion models and diverse 3D data (Deitke et al., 2023). Zero123 frames NVS as a viewpoint-conditioned image-to-image translation problem. It leverages a pre-trained image translation model and fine-tunes on multi-view renderings of 3D dataset. However, the image-to-image translation model lacks high-level semantic guidance to constrain the NVS solution space as illustrated in Fig. 1 (a-b), often resulting in implausible results.

**3D Generation with Diffusion Models.** Drawing inspiration from 2D generation, recent endeavours have been made to train generative models on 3D datasets conditioned on text or images (Cheng et al., 2023; Liu et al., 2023d; Gupta et al., 2023; Zheng et al., 2023). However, they are usually confined to specific categories with unsatisfactory generation results. On the other end, optimization-based 3D generation (Poole et al., 2022; Lin et al., 2023; Deng et al., 2023) can distill knowledge of pre-trained text-to-image diffusion models to a 3D model given any text. However, due to limited camera control over 2D pre-trained models, they often suffer from blurriness and the Janus problems, with low generation success rates. Subsequent research has mitigated the problem with additional priors such as condition images (Melas-Kyriazi et al., 2023; Tang et al., 2023), depth (Seo et al., 2023), and human body skeletons (Huang et al., 2023b). Zero123 (Liu et al., 2023b) empowers a pre-trained image-to-image diffusion model with NVS capabilities, which leads to more accurate control over camera poses of generated images. However, due to the limitation of NVS quality, the 3D generation results still remain sub-optimal.

**Remark.** TOSS *lies in the intersection of the above three directions.* It utilizes a 2D Stable Diffusion model to target at the zero-shot open-set single-view NVS task. The capability of generating semantically plausible multiview-consistent images enables TOSS to be readily integrated in various downstream applications, such as 3D generation and reconstruction.

# 3 METHOD

We aim at solving the issue of NVS models being under-constrained with text guidance. Here we provide Fig. 1 (a-b) to exemplify the problem and motivate our approach. We start with a discussion on some preliminary knowledge in section 3.1. To effectively incorporate text information into our framework, we make two efforts. First, in section 3.2 we describe our novel text-conditioned model formulation and discuss the advantages of employing textual guidance. Subsequently, section 3.3 elaborates on our novel mechanisms to enable text-to-image diffusion models to condition on images and camera poses, so that they can fit into our desired model formulation and be used for the NVS task. Fig. 2 illustrates the overall pipeline of our proposed method TOSS.

## 3.1 PRELIMINARY

**Diffusion Models** consist of two processes: the forward process of adding noise and the reverse process of denoising. In the forward process, a noisy image $\mathbf{x}_t$ is obtained by adding a random Gaussian noise $\boldsymbol{\epsilon} \sim \mathcal{N}(0, I)$ to a clean image $\mathbf{x} \sim p(\mathbf{x})$ according to the weights $\bar{\alpha}_t$ determined by timestep $t \in [0, 1000]$, while in the reverse process, a trainable U-Net $\boldsymbol{\epsilon}_\theta$ predicts noise and denoises images at different timesteps: $\min_\theta \ \mathbb{E}_{t,\mathbf{x},\boldsymbol{\epsilon}}\left[\|\boldsymbol{\epsilon} - \boldsymbol{\epsilon}_\theta(\mathbf{x}_t, t)\|_2^2\right]$. After training, one can generate an image $\hat{\mathbf{x}} \sim p(\mathbf{x})$ by starting from an isotropic Gaussian noise and gradually denoising it.

**Single-view Novel View Synthesis**. Given a condition image $\mathbf{x}_c$ of an object, the single-view NVS task is to synthesize an image of the object from another camera viewpoint. Let $P$ denote the camera transformation, i.e., rotation and translation, of the desired viewpoint relative to the viewpoint of $\mathbf{x}_c$. The single-view NVS problem can be formulated as $\hat{\mathbf{x}} = f(\mathbf{x}_c, P)$, where we intend to learn a function $f(\cdot, \cdot)$ for conducting NVS.

## 3.2 TOSS: TEXT-GUIDED NOVEL VIEW SYNTHESIS

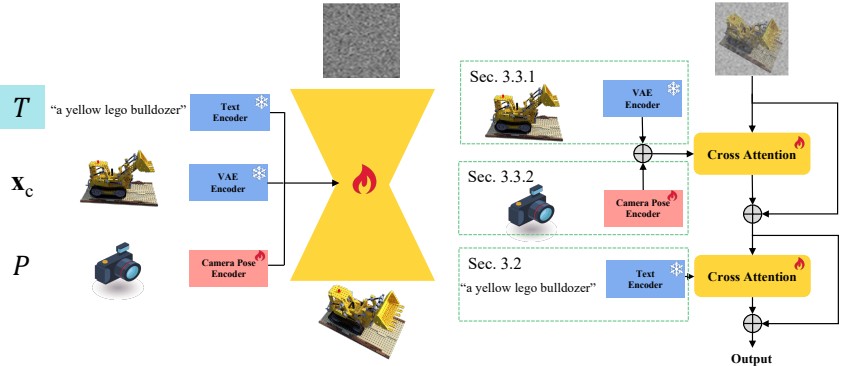

Figure 2: **The pipeline of TOSS (Left) and our conditioning mechanisms (Right)**.

Formally, the model formulation of TOSS differs from previous works by integrating text descriptions into the novel-view synthesis process. *TOSS* is modeled as:

$$\hat{\mathbf{x}} = f(\mathbf{x}_c, P, \boxed{T}), \tag{1}$$

where $\boxed{T}$ denotes the textual description that aligns with $\mathbf{x}_c$. As shown in Fig. 2 (Right), the text information is first encoded by a frozen text encoder (CLIP) and then integrated into the novel view generation network through a cross-attention mechanism.

Compared to the original NVS, TOSS further constrains the solution space of NVS with extra text information, demonstrating two substantial advantages:

- **Increasing Plausibility.** Text can serve as high-level semantic information that directs NVS generation toward a more plausible solution that is consistent with the text description. As shown in Fig. 1 (a), being aware that the input image is a shark will ensure that the synthesized novel view should not have two tails.

- **Increasing Controllability.** When generating unobservable parts of an object, text can impose constraints on the solution space where multiple syntheses may be plausible. As shown in Fig. 1 (b), users could explicitly specify the content (*e.g.* tap or flowers) to be synthesized within the concealed part of the input image.

Since the problem of novel view synthesis from a single-view image is severely under-constrained, we formulate $f$ by adapting a text-to-image diffusion model to condition on both the input view image and the camera pose, in order to account for the non-deterministic nature of the synthesis process. Formally, our $\epsilon_\theta$ takes the form as $\epsilon_\theta(\mathbf{x}_t, t, \mathbf{x}_c, P, T)$, and it is trained as follows:

$$\min_\theta \mathbb{E}_{t,\mathbf{x},\epsilon,\mathbf{x}_c,P}\left[\left\|\epsilon - \epsilon_\theta(\mathbf{x}_t, t, \mathbf{x}_c, P, T)\right\|_2^2\right]. \tag{2}$$

For sampling, we employ classifier-free guidance for both condition image and text, and set two guidance scales $(\alpha, \beta)$ to control their influence respectively. Since these two conditions are not independent (Liu et al., 2022), we borrow inspiration from (Brooks et al., 2023) and employ a composed score function $\hat{\epsilon}_\theta$ as follows:

$$\begin{aligned}
\hat{\epsilon}_\theta(\mathbf{x}_t, t, \mathbf{x}_c, P, T) = &\ \epsilon_\theta(\mathbf{x}_t, t, \emptyset, P, \emptyset) \\
&+ \alpha[\epsilon_\theta(\mathbf{x}_t, t, \mathbf{x}_c, P, \emptyset) - \epsilon_\theta(\mathbf{x}_t, t, \emptyset, P, \emptyset)] \\
&+ \beta[\epsilon_\theta(\mathbf{x}_t, t, \mathbf{x}_c, P, T) - \epsilon_\theta(\mathbf{x}_t, t, \mathbf{x}_c, P, \emptyset)],
\end{aligned} \tag{3}$$

where we slightly abuse our notation and let $\emptyset$ represent the null symbol for both conditions.

Note that TOSS is a highly flexible framework. Even in the absence of text descriptions, which means that TOSS degenerates to a classical NVS model $f(\mathbf{x}_c, P)$, it consistently generates high-quality target images. We will demonstrate this ability in our ablation study.

## 3.3 ADAPTING A TEXT-TO-IMAGE MODEL FOR NVS

Recall that TOSS is formulated as $\hat{\mathbf{x}} = f(\mathbf{x}_c, P, T)$. Practically, in TOSS we capitalize on Stable Diffusion (Rombach et al., 2022), whose formulation is $\hat{\mathbf{x}} = f_{SD}(T)$, due to its remarkable ability to generate diverse images according to text descriptions. While retaining Stable Diffusion's text-conditioning mechanism, we still need to enable it to condition on the input view image $\mathbf{x}_c$ and the relative camera pose $P$ for the task of NVS. Hence in the following subsections, we will describe:

- A dense cross attention module to condition $f_{SD}$ on input view images. Introducing this mechanism to $f_{SD}$ gives us a model $g$ of the form $g(\mathbf{x}_c, T)$.

- Our mechanism for merging camera poses into the cross-attention module of $g$, to reach to our desired function, namely TOSS, $f(\mathbf{x}_c, P, T)$ that is conditioned on condition images, camera poses, and text descriptions.

### 3.3.1 ENABLING TEXT-TO-IMAGE MODELS TO CONDITION ON IMAGE

Given an input view image $\mathbf{x}_c$, Zero123 (Liu et al., 2023b) utilizes two image conditioning mechanisms as illustrated in Fig. 3: (a) concatenating features of input image $\mathbf{x}_c$ with $\mathbf{x}_t$ along the channel dimension; (b) cross attention with $\mathbf{x}_c$'s CLIP embedding. However, both mechanisms possess inherent limitations and are inadequate for the novel view synthesis task.

The underlying reasons for the inadequacy of previous conditioning mechanisms can be attributed to information misalignment and excessive information compression. *Information misalignment:* the feature maps produced by the VAE encoder of Stable Diffusion indicate not only the existence of certain local features but also their positions in the 2D image. Thus, when image features are concatenated for NVS as shown in Fig. 3 (a), the aforementioned property implicitly requires the feature maps of $\mathbf{x}_c$ and $\mathbf{x}_t$ to be aligned spatially. Namely, $\mathbf{x}_c$ and $\mathbf{x}_t$ must have alike object poses, which is hardly met in real-world NVS scenarios. Therefore the concatenation method is inappropriate for learning features of novel poses. *Excessive information compression:* Zero123 utilizes features of $\mathbf{x}_t$ as queries and CLIP embeddings of $\mathbf{x}_c$ as keys and values in the cross-attention mechanism. However, since CLIP contracts most of the local features and embeds the whole image into a single

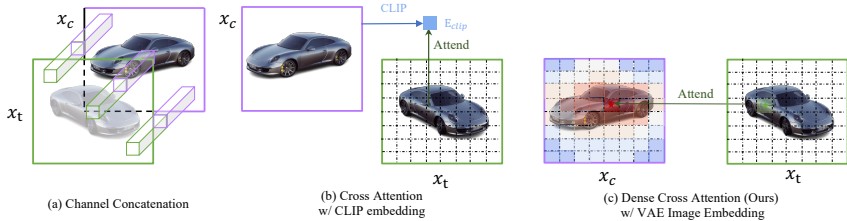


(a) Channel Concatenation      (b) Cross Attention
w/ CLIP embedding      (c) Dense Cross Attention (Ours)
w/ VAE Image Embedding


Figure 3: **Comparing previous image conditioning mechanisms (a-b) and TOSS (c).**

token, the cross attention module of Zero123 is actually incapable of attending to local features of the underlying object to be synthesized.

In contrast to previous approaches, we conduct cross attention between VAE embeddings of $\mathbf{x}_t$ and $\mathbf{x}_c$, in preservation of $\mathbf{x}_c$'s local details. Let $\boldsymbol{f}_{\mathbf{x}_t}$, $\boldsymbol{f}_{\mathbf{x}_c} \in \mathbb{R}^{h \times w \times d}$ indicate the aforementioned VAE embeddings respectively, where $h$, $w$, and $d$ denote the height, width and channels of the feature maps. We linearly project $\boldsymbol{f}_t$ and $\boldsymbol{f}_c$ into query, key, value features $\mathbf{Q}_{\mathbf{x}_t}, \mathbf{K}_{\mathbf{x}_c}, \mathbf{V}_{\mathbf{x}_c} \in \mathbb{R}^{n \times d}$, where $n = h \times w$, and perform standard cross attention.

Compared to the concatenation method, in our setting any feature from $\mathbf{x}_t$ could freely attend to any feature from $\mathbf{x}_c$, unconstrained by their pose difference. Compared to the coarse attention with CLIP embeddings, as the features from $\mathbf{x}_c$ and $\mathbf{x}_t$ pass through the downsampling modules of our $\epsilon_\theta$, the information granularity gradually increases. Hence the cross attention module takes into consideration information from both global and local levels.

We summarize the three conditioning mechanisms as below:

- **Channel Concatenation**: output $= \boldsymbol{f}_{\mathbf{x}_t} \oplus \boldsymbol{f}_{\mathbf{x}_d} : (\mathbb{R}^{n \times d}, \mathbb{R}^{n \times d}) \to \mathbb{R}^{n \times 2d}$
- **Cross Attention**: output $= \text{Cross-Attn}(\mathbf{Q}_{\mathbf{x}_t}, \mathbf{K}_{\mathbf{E}_{clip}}, \mathbf{V}_{\mathbf{E}_{clip}}) : (\mathbb{R}^{n \times d}, \mathbb{R}^{1 \times d}, \mathbb{R}^{1 \times d}) \to \mathbb{R}^{n \times d}$
- **Ours**: output $= \text{Cross-Attn}(\mathbf{Q}_{\mathbf{x}_t}, \mathbf{K}_{\mathbf{x}_c}, \mathbf{V}_{\mathbf{x}_c}) : (\mathbb{R}^{n \times d}, \mathbb{R}^{n \times d}, \mathbb{R}^{n \times d}) \to \mathbb{R}^{n \times d}$

### 3.3.2 ENABLING TEXT-TO-IMAGE MODELS TO CONDITION ON RELATIVE CAMERA POSE

The relative camera pose is extremely important for modulating the correspondence between the input view and the target view for NVS. However, although being conditioned on texts, Stable Diffusion does not depend on camera poses, making it not directly applicable to the NVS task. Therefore, in this paper, we introduce a simple yet effective pose conditioning mechanism that reflects such modulation effects.

Ideally, the relative camera pose should provide informative guidance on feature querying : *e.g.* if $\mathbf{x}_t$ and $\mathbf{x}_c$ share the same viewpoint, where the relative camera pose is zero, the querying attention should concentrate more on keys from corresponding positions. Therefore we transform the camera pose to an extra key-value pair $\mathbf{K}_P, \mathbf{V}_P \in \mathbb{R}^{1 \times d}$ via learned projection matrices, append them to $\mathbf{K}_{\mathbf{x}_c}$ and $\mathbf{V}_{\mathbf{x}_c}$ respectively as $\mathbf{K}_{\mathbf{x}_c, P} = [\mathbf{K}_{\mathbf{x}_c}; \mathbf{K}_P]$ and $\mathbf{V}_{\mathbf{x}_c, P} = [\mathbf{V}_{\mathbf{x}_c}; \mathbf{V}_P]$, where $\mathbf{K}_{\mathbf{x}_c, P}$ and $\mathbf{V}_{\mathbf{x}_c, P} \in \mathbb{R}^{(n+1) \times d}$. The pose-augmented key and value features are then used in the cross attention module as shown in Fig. 2. The introduction of a new pose key-value pair alters the distribution of its attention coefficients, thereby implicitly modulating the attention strength between $\mathbf{x}_t$ and $\mathbf{x}_c$. In contrast to previous works (Watson et al., 2022) (Liu et al., 2023b), we additionally perform frequency encoding on the camera pose so that the pose key-value pair preserves more high-frequency information (Mildenhall et al., 2021).

**More Accurate Pose via Expert Denoisers.** Concurred with recent analyses on diffusion process (Huang et al., 2023a; Choi et al., 2022; Balaji et al., 2022), we observe that during sampling, $\epsilon_\theta$ at large timesteps dictates novel view pose and further refines the NVS details at small timesteps. See illustrations in App. C.3. Drawing inspiration from ensemble of expert denoisers (Balaji et al., 2022), we train two expert denoisers that are specialized for denoising at different timestep intervals, responsible for pose correctness and fine details respectively during the generative process. In order to boost NVS quality while maintaining the same training and inference computation cost, we begin by training a model shared among all noise levels. Then we initialize two expert models from this base model and specialize them to high (timestep 1000-800) and low (timestep 800-0) noise levels.

# 4 EXPERIMENTS

We evaluate TOSS on the tasks of single-view NVS and 3D reconstruction. Note that the 3D reconstruction task is more challenging, and demands higher multiview consistency. In Sec. 4.1, we describe our data collating and model training protocols. The results on single-view NVS and 3D reconstruction are presented in Sec. 4.2 and Sec. 4.3 respectively. Ablations on the design choices are provided in Sec. 4.4. Finally, we highlight in Sec. 4.5 that our improvements can be readily combined with upgrades in text-to-image models and techniques for even higher-quality NVS. All the visualizations provided in this section are produced without the two-stage expert denoisers.

## 4.1 EXPERIMENTAL SETTINGS

**Captioning 3D Data.** We employ an automated captioning procedure similar to that of Cap3D (Luo et al., 2023) but alter its final-stage caption fusion strategy slightly to caption Objaverse (Deitke et al., 2023), the currently largest and most diverse, but captionless, 3D dataset. The detailed captioning procedure can be found in App. C.1. We integrate Cap3D's 660K 3D-text pairs with 140K self-captioned 3D objects, resulting in a total of 800K Objaverse 3D-text pairs for training TOSS.

**Implementation Details.** We train TOSS on captioned 3D data as described above. For each 3D instance, we randomly sample 12 views for training. As shown in Fig. 2, we initialize TOSS with pre-trained Stable Diffusion v1.5 with both CLIP encoder and VAE encoder frozen. We train 120K steps with a total batch size of 2048 on 8 A100 GPUs which takes about 7 days. For training with expert denoisers, we first train 60K steps for all noise levels, then initialize two expert models from this base model and resume training for 12K and 48K steps respectively for high (timestep 1000-800) and low (timestep 800-0) noise levels.

## 4.2 NOVEL VIEW SYNTHESIS FROM A SINGLE IMAGE

**Quantitative Results.** Following Zero123 Liu et al. (2023b), we evaluate on GSO (Downs et al., 2022) and RTMV (Tremblay et al., 2022) for **NVS quality** in Tab. 1 , each with 20 randomly selected objects/scenes rendered for 17 randomly sampled views, which are not included in the training dataset. We use four standard metrics for evaluation: PSNR, SSIM (Wang et al., 2004), LPIPS (Zhang et al., 2018) and KID. We keep the number of training images the same as the baseline (Zero123) for a fair comparison.

| Method | Training images | Google Scanned Objects (GSO) | | | | RTMV | | | |
|---|---|---|---|---|---|---|---|---|---|
| | | PSNR (↑) | SSIM (↑) | LPIPS (↓) | KID (↓) | PSNR (↑) | SSIM (↑) | LPIPS (↓) | KID (↓) |
| Image Variation | – | 10.33 | 0.3094 | 0.3618 | 0.0543 | – | – | – | – |
| Diet-NeRF | – | 12.34 | 0.3290 | 0.4611 | 0.1211 | – | – | – | – |
| Zero1-to-3 | 160M | 17.75 | 0.8139 | 0.1369 | 0.0046 | 9.58 | 0.4180 | 0.3845 | 0.0267 |
| TOSS (inference w/o text) | 160M | 18.45 | 0.8401 | 0.1231 | 0.0046 | 10.50 | 0.5080 | 0.3497 | 0.0147 |
| TOSS (inference w/ text) | 160M | 19.49 | 0.8580 | 0.1142 | **0.0036** | 10.75 | 0.5187 | 0.3360 | **0.0128** |
| TOSS (w/ expert denoisers) | 160M | **19.70** | **0.8589** | **0.1131** | 0.0027 | **11.22** | **0.5823** | **0.3353** | 0.0132 |
| Zero1-to-3 | 250M | 18.67 | 0.8322 | 0.1257 | **0.0023** | 10.28 | 0.4867 | 0.3592 | 0.0156 |
| TOSS (inference w/o text) | 250M | 19.91 | 0.8649 | 0.1116 | 0.0034 | 11.39 | 0.5660 | 0.3213 | 0.0130 |
| TOSS (inference w/ text) | 250M | 20.09 | 0.8685 | 0.1114 | 0.0032 | 11.54 | 0.5734 | 0.3139 | 0.0119 |
| TOSS (w/ expert denoisers) | 250M | **20.16** | **0.8693** | **0.1109** | 0.0032 | **11.62** | **0.5754** | **0.3115** | **0.0104** |

Table 1: **Quantitative comparison of single-view novel view synthesis on GSO and RTMV.**

Furthermore, we evaluate the **3D consistency scores** following (Watson et al., 2022). Specifically, we randomly sample 100 camera poses in the whole sphere, with 80 used for training and 20 for testing. Given a single image, different methods generate novel views based on the 100 camera poses according to the input view image. Then we

| Method | GSO | | | RTMV | | |
|---|---|---|---|---|---|---|
| | PSNR (↑) | SSIM (↑) | LPIPS (↓) | PSNR (↑) | SSIM (↑) | LPIPS (↓) |
| Zero123 | 21.05 | 0.8893 | 0.2754 | 11.38 | 0.4350 | 0.6420 |
| TOSS(inference w/ text) | **21.54** | **0.8903** | **0.2700** | **12.36** | **0.4696** | **0.6186** |

Table 2: **Comparing 3D consistency scores on GSO and RTMV.**

fit a NeRF model on the generated novel-view images, and evaluate the NeRF model with PSNR, SSIM and LPIPS over the 20 testing images. As shown in Tab. 2, our model consistently improves multi-view consistency amongst different generated views.

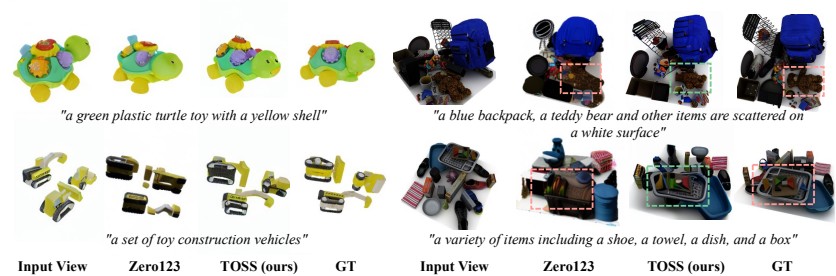

Figure 4: **Qualitative comparison of single-view NVS, on GSO (Left) and RTMV (Right).**

**Qualitative Results.** We show that TOSS consistently generates higher-quality novel view images compared to Zero123, evaluated across GSO (Fig. 4 Left), RTMV (Fig. 4 Right) and in-the-wild web images (Fig. 1 (c-d)). Fig. 1 (a,b) shows that TOSS produces more plausible and controllable generations, respectively. Furthermore, it can be observed in Fig. 1 (c) that TOSS achieves significantly higher multiview-consistency across multiple generations, using the same textual description.

## 4.3 3D RECONSTRUCTION

We evaluate TOSS for 3D reconstruction on GSO and RTMV instances, as well as randomly selected web images, with corresponding captions obtained via BLIP2 (Li et al., 2023a). We use the SDS (Poole et al., 2022) loss implemented by threestudio Guo et al. (2023) for 3D reconstruction, evaluated both qualitatively, and quantitatively on exported mesh from reconstructed 3D NeRFs.

**Quantitative Results.** To evaluate 3D reconstruction, we report the commonly used Chamefer Distances (CD) and Volume IoU between ground truth meshes and reconstructed meshes. Following previous work, we randomly select 20 instances each for GSO and RTMV. For each instance, we render an image with resolution $256 \times 256$ as the input view. The numerical

| Method | GSO | | RTMV | |
|---|---|---|---|---|
| | CD ($\downarrow$) | IoU ($\uparrow$) | CD ($\downarrow$) | IoU ($\uparrow$) |
| Magic123 | 0.1508 | 0.3220 | – | – |
| Zero123 | 0.0757 | 0.5151 | 0.1546 | 0.2037 |
| TOSS(inference w/ text) | **0.0655** | **0.5205** | **0.1236** | **0.2136** |

Table 3: **Quantitative comparison of single-view 3D reconstruction on GSO and RTMV.**

results in Tab. 3 demonstrate that TOSS outperforms Zero123 in terms of reconstruction accuracy.

**Qualitative Results.** Fig 5 shows that TOSS reconstructs higher-fidelity 3D models than baseline method Zero123, with finer details and better 3D meshes, given just a single image.

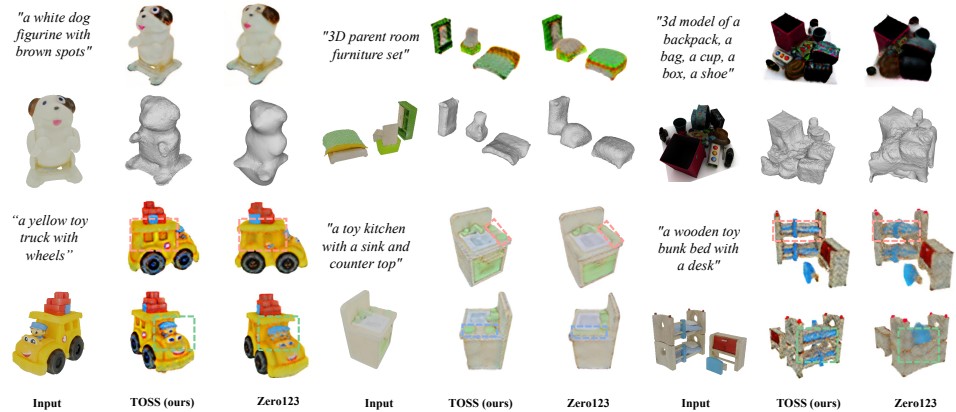

Figure 5: **Qualitative comparison of 3D reconstruction on GSO and RTMV.**

## 4.4 ABLATION STUDY

We present in Tab. 4 ablations on the contributions of different design choices of TOSS. For more ablations please refer to App. D. We ablate on a subset of Objaverse, dubbed *Objaverse vehicles*, constituted of instances from the "vehicles" category including cars, ships, trains, spaceships and etc. The subset contains 3,000 instances with 12 randomly sampled views for training, and 20 instances with 16 randomly sampled views for testing. The subset is constructed to be both efficient for

training (comparing to the full Objaverse) and contains enough diversity for evaluation (comparing to ShapeNet). For each ablation, we train on Objaverse vehicles on a total number of 450K images with a batch size of 1024.

We start from Zero123 and progressively ablate on the contributions of the following changes: 1) replace the original image conditioning mechanisms with our dense cross attention as detailed in 3.3.1; 2) condition on text by replacing the CLIP image encoder with text encoder, initialized with pre-trained weights from Stable Diffusion; 3) con-

| Modules | Contribution | | Metrics | |
|---|---|---|---|---|
| | PSNR (↑) | SSIM (↑) | PSNR (↑) | SSIM (↑) |
| Zero123 | - | - | 12.73 | 0.6123 |
| + Token-level attention | +2.70 | +0.1762 | 15.43 | 0.7885 |
| + Text prompt | +1.41 | +0.0390 | 16.84 | 0.8275 |
| + Camera pose token | +0.11 | +0.0094 | 16.95 | 0.8369 |

Table 4: **Module ablations on Objaverse vehicles.**

dition on camera pose with dense cross attention instead of concatenation with CLIP embedding as detailed in 3.3.2. We observe that all introduced designs contribute positively to the NVS task, where dense cross attention and text conditioning attains the most significant improvements.

## 4.5 Integration with Higher-capability Text-to-Image Models and Techniques

**Higher-capability Text-to-image Model.** We replace the SD v1.4 model in TOSS with the higher-resolution (256 to 512) SD v1.5, and evaluate for NVS. As shown in Fig. 6 (Left), TOSS-512 not only generates finer details, but also aligns its generations better with the text descriptions.

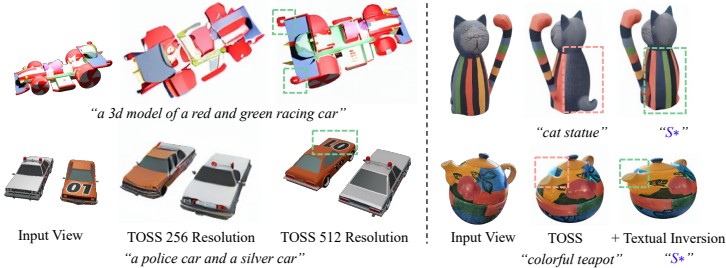

Figure 6: **Higher-quality NVS with TOSS.** (Left) TOSS produces much higher quality NVS images with higher-capability generative models. (Right) TOSS further improves NVS quality by optimizing a $S_*$ as object caption via textual inversion (Gal et al., 2022). Input texts are augmented with shared prefix "a photo of " and fine details are highlighted with dashed rectangles.

**Captioning Input View Image with Textual Inversion.** Describing a specific object precisely can be challenging. In such cases, we can leverage techniques from text-to-image concept generation, such as textual inversion (Gal et al., 2022). To this end, we learn a unique text embedding $S_*$ for the input view image and use it for subsequent NVS with TOSS. As depicted in 6 (Right), textual inversion further enhances the consistency with the input view (*e.g.*, for the "cat statue" example, it produces more plausible color stripes on the back and avoids adding an extra illusionary tail).

## 5 Limitations and Conclusion

**Limitations and Future Works.** Comparing to traditional NVS, TOSS optionally takes an extra textual description, which can be obtained automatically via captioning models. Developing captioning models more suited to describing object details for NVS remains for future work. Similar to Zero123, we fine-tune Stable Diffusion on synthetic 3D datasets, which is prone to training distribution shift, utilizing natural images and videos for training would alleviate this problem.

**Conclusions.** In this paper, we have presented TOSS to generate more plausible, controllable, and multiview-consistent novel view images from a single image. TOSS utilizes text as semantic guidance to further constrain the solution space of NVS. We naturally capitalize on Stable Diffusion and introduce novel dense cross attention module to condition the novel view generative process on input view image and relative camera pose. We also propose using expert denoisers to further enhance novel view pose correctness and fine details. Extensive experiments demonstrate that TOSS generates novel view images that are significantly more plausible, controllable, and multiview-consistent than Zero123on NVS and 3D reconstruction tasks. Furthermore, we showcase the substantial potential inherent to our approach by utilizing higher-capability text-to-image models and techniques to further boost NVS quality.

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

## A    INTRODUCING TEXT TO CONSTRAIN THE NVS SOLUTION SPACE

Current NVS methods (Liu et al., 2023b; Chan et al., 2023) generate novel view images conditioned on input view images and camera poses, resulting in a solution space highlighted in pink as shown in Fig. 7. TOSS further constrains the novel-view generation process with text as high-level semantics, which yields a more compact solution space as highlighted in green . Plausible image generation results conditioned on each of the modalities (image, camera pose, text) are presented for reference. Notably, image-conditioned generation is widely known as image-to-image variation which focuses on producing images that roughly "look like" the input images (in this case red-colored standing characters in the lower left part). On the other end, text-conditioned image generation represented by Stable Diffusion (Rombach et al., 2022) emphasizes semantically correct generations of "Iron Man" in different poses.

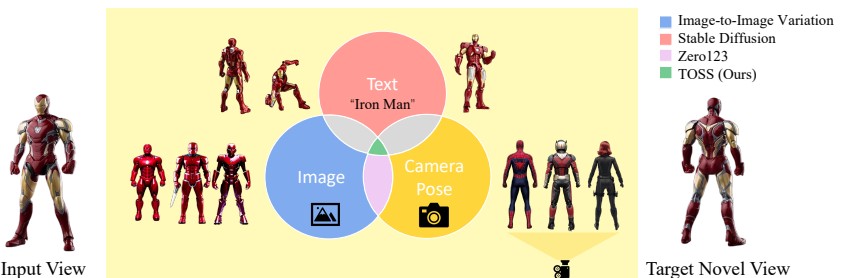

Figure 7: **An overview of utilizing different conditions for image generation.** For each condition, three plausible solutions obeying the respective constraint are provided for illustration. While Zero123 considers the input view image and camera pose, TOSS further introduces text as semantic constraint for generating more plausible and controllable novel view synthesis (NVS) result.

## B    CONCISE PIPELINE FOR NVS

Other than the capability of utilizing text to enable more plausible and controllable NVS generation, TOSS also employs a more concise training pipeline as shown in Fig. 8. Zero123 (Liu et al., 2023b) requires fine-tuning Stable Diffusion on large-scale image collections to convert the original text-to-image model to an image-to-image one, which is both resource demanding and subject to training distribution shift. TOSS completely skips this problematic step, making our improvements fully orthogonal to and can be readily combined with upgrades in foundation text-to-image models, *e.g.* better architectures, textual inversion (Gal et al., 2022), *etc.*, to boost the performance further.

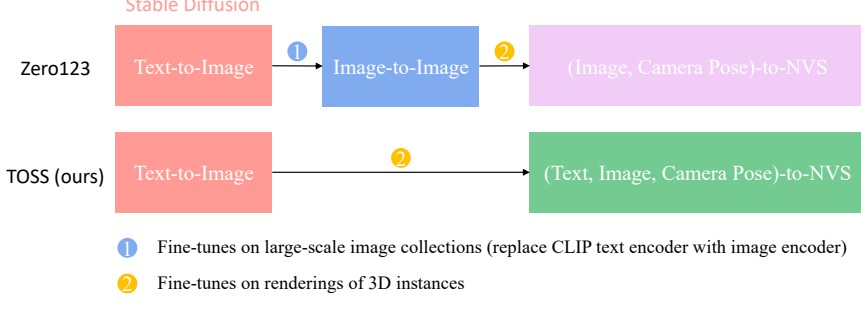

Figure 8: **TOSS utilizes a more concise training pipeline.**

## C    MORE EXPERIMENTAL SETTINGS

### C.1    MORE DETAILS ON CAPTIONING DATA

We utilize an automated pipeline to caption 3D data. Specifically, we first employ BLIP2 (Li et al., 2023a) to caption a collection of 3D object renderings. Subsequently, we apply a CLIP-based (Radford et al., 2021) ranking mechanism to mitigate potential inconsistencies that may arise when generating captions for different viewpoints of the same object. Lastly, we incorporate several regular expression rules to filter out inaccurate descriptions. Concurrently, we observe that Cap3D (Luo et al., 2023) follows a similar procedure but utilizes GPT-4 (OpenAI, 2023) at the final stage to fuse filtered captions across different views.

At inference time, users can either provide no text information or use image captioning models such as BLIP2 (Li et al., 2023a) for automatic captioning.

### C.2    MORE TRAINING DETAILS

During model training we employ an AdamW optimizer (Loshchilov & Hutter, 2017) with a learning rate of $10^{-3}$ for the camera pose encoder and $10^{-4}$ for other modules. In all the experiments, we train our model with the 16-bit floating point (fp16) format for efficiency. To enable classifier-free guidance, we randomly mask 50% of the samples for text in each batch and 10% for condition images. For computing the consistency score, we directly train an instant-ngp (Müller et al., 2022) for 30 epochs on the generated novel views. For 3D generation we adopt the SDS loss (Poole et al., 2022) and hyper-parameter settings implemented by threestudio (Guo et al., 2023). The instant-ngp model first renders novel-view images at a resolution of $64 \times 64$, which are then up-sampled to $256 \times 256$ resolution and supervised in the latent space. It takes about 3 minutes for training 400 epochs with a batch size of 12. After model training, we apply the marching cubes algorithm (Lorensen & Cline, 1998) to export meshes from the trained instant-ngp.

### C.3    EXPERT DENOISERS

We visualize the novel view image generation process in Fig. 9. For large timesteps $t$, the gradients provided by $\epsilon_\theta$ supervise the geometry while small timesteps induce gradients that focus more on fine details. The observation motivates us to train dual expert denoisers for novel view pose correctness and fine details respectively as described in the main paper.

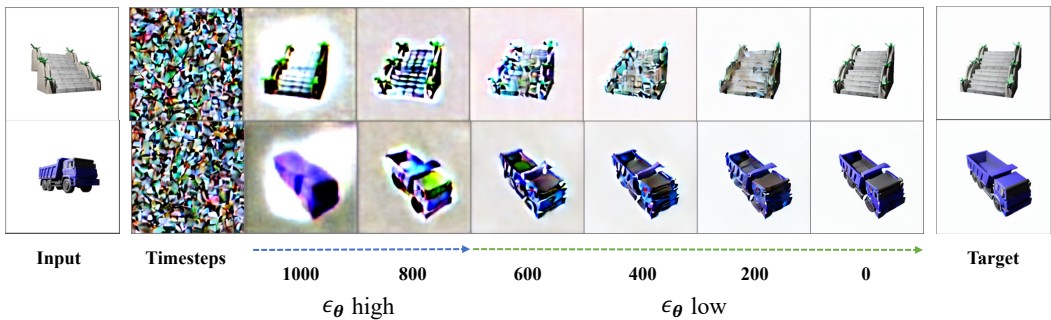

Figure 9: **The generative process of NVS.** $\epsilon_\theta$ at large timesteps ensure the accuracy of novel view poses and small timesteps are responsible for generating fine details.

# D   MORE ABLATION STUDY

**Guidance Settings**   Given that we employ classifier-free guidance (CFG) for both conditional images and texts, the need of trade-off between their respective guidance scales naturally arises. Hence we have conducted ablations on GSO and RTMV datasets, to explore various combinations of guidance scales as illustrated in Fig. 10. It is worth emphasizing that the dual-guidance configuration offers flexibility to cater to different scenarios. In conclusion, we suggest raising the text guidance scale, $\beta$, when prioritizing the text controllability of TOSS. Conversely, a higher image guidance scale, $\alpha$, is recommended when the provided condition images contain sufficient information on the objects.

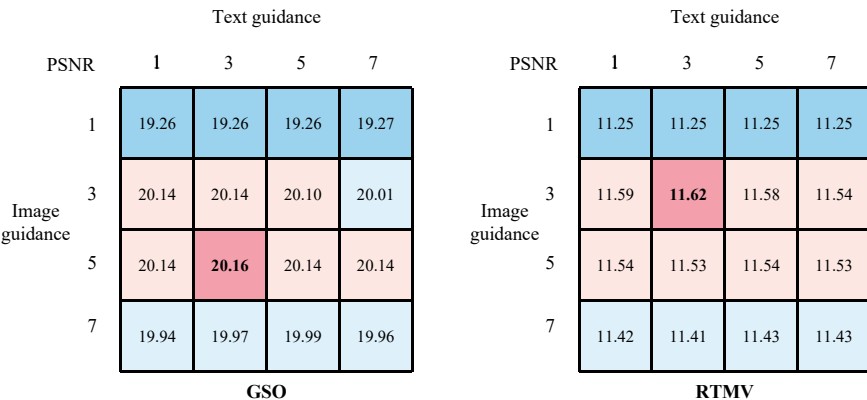

Figure 10: **Performance of different guidance settings on GSO and RTMV.**

**Quality of Texts**   TOSS aims to utilize text as high-level semantics to constrain the NVS solution space. Consequently, the quality of the given text plays a significant role in influencing the quality of the generated results. Here we delve into the impact of text descriptions at varying levels of granularity on the results. From Figure 11, it is evident that when provided with a coarse text, there may exist slight deviations in the generated object details (for instance, a change in the color of the unicorn's horn). However, by further refining the granularity of the prompt description, these deviations can be avoided with more precise textual control.

Furthermore, we conduct manual re-annotation with more detailed descriptions on the GSO subset. This text refinement leads to an increase of 0.1 in the PSNR value compared to the initial assessment.

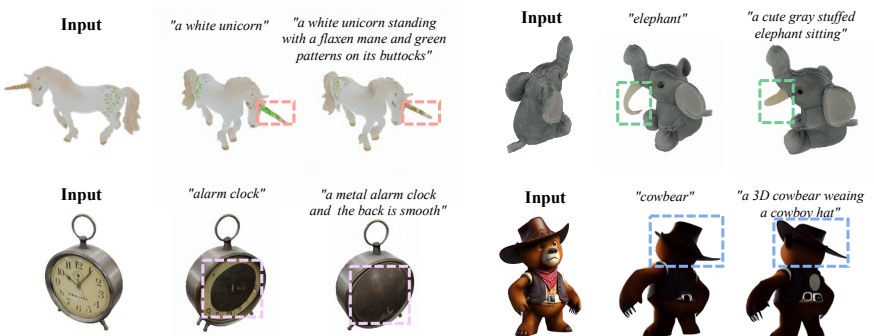

Figure 11: **Refining text prompts enhances the quality of generated novel views.**

**Importance of Using Pre-trained Weights.**   During our experiments, we also explore training TOSS from scratch rather than initializing TOSS with pre-trained weights of Stable Diffusion. Our experiments are conducted on the Objaverse vehicles dataset as detailed in the main paper Sec. 4.4, with the same number of training steps. The comparative analysis of the novel view synthesis results

is presented in Fig. 12. It is evident that in the absence of text-to-image pre-training, the novel view generations exhibit significant distortions in shape and texture, validating the importance of utilizing pre-trained knowledge from generative models.

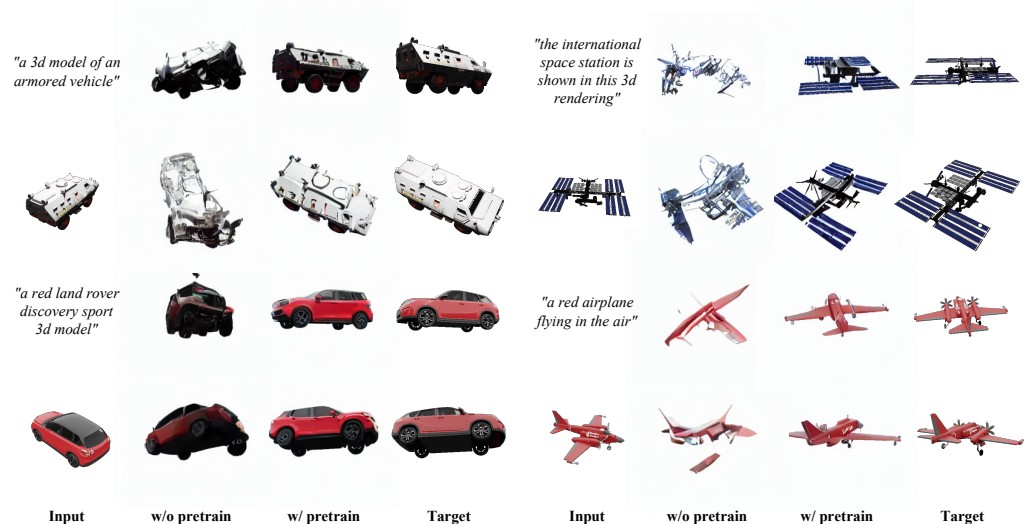

Figure 12: **Importance of initializing TOSS with pre-trained Stable Diffusion.**

# E   QUALITATIVE COMPARISON FOR CONTRIBUTIONS OF DIFFERENT MODULES

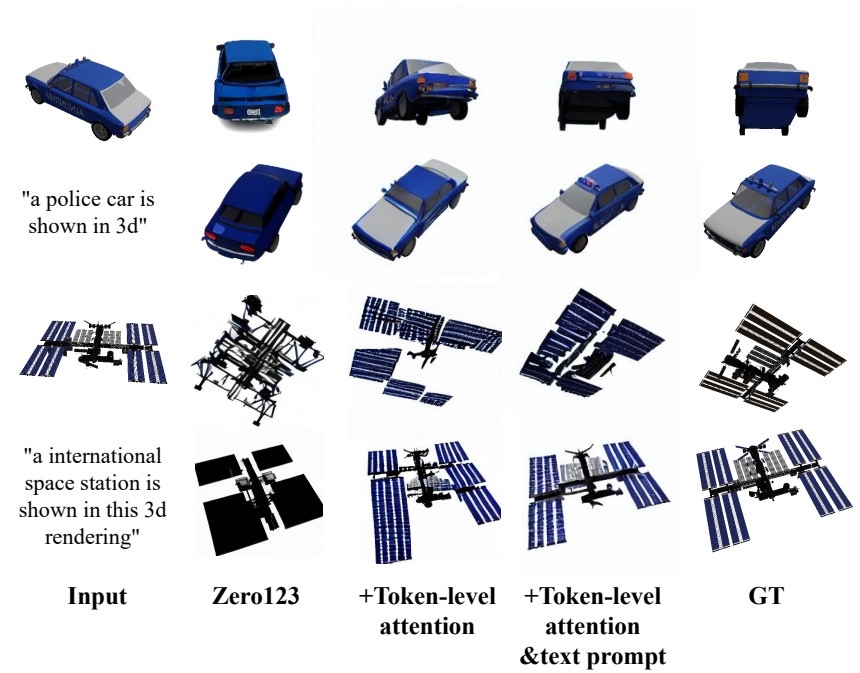

Figure 13: **Qualitative comparison for contributions of different modules**.

# F   MORE PLAUSIBILITY RESULTS

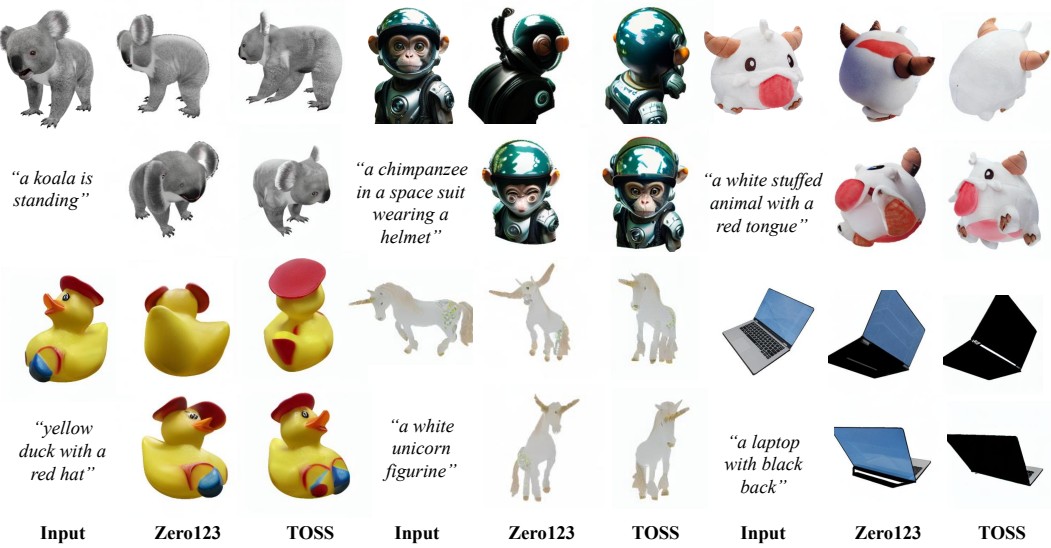

Figure 14: **More results demonstrating the increased NVS plausibility with TOSS.**.

## G    MORE CONTROLLABILITY RESULTS

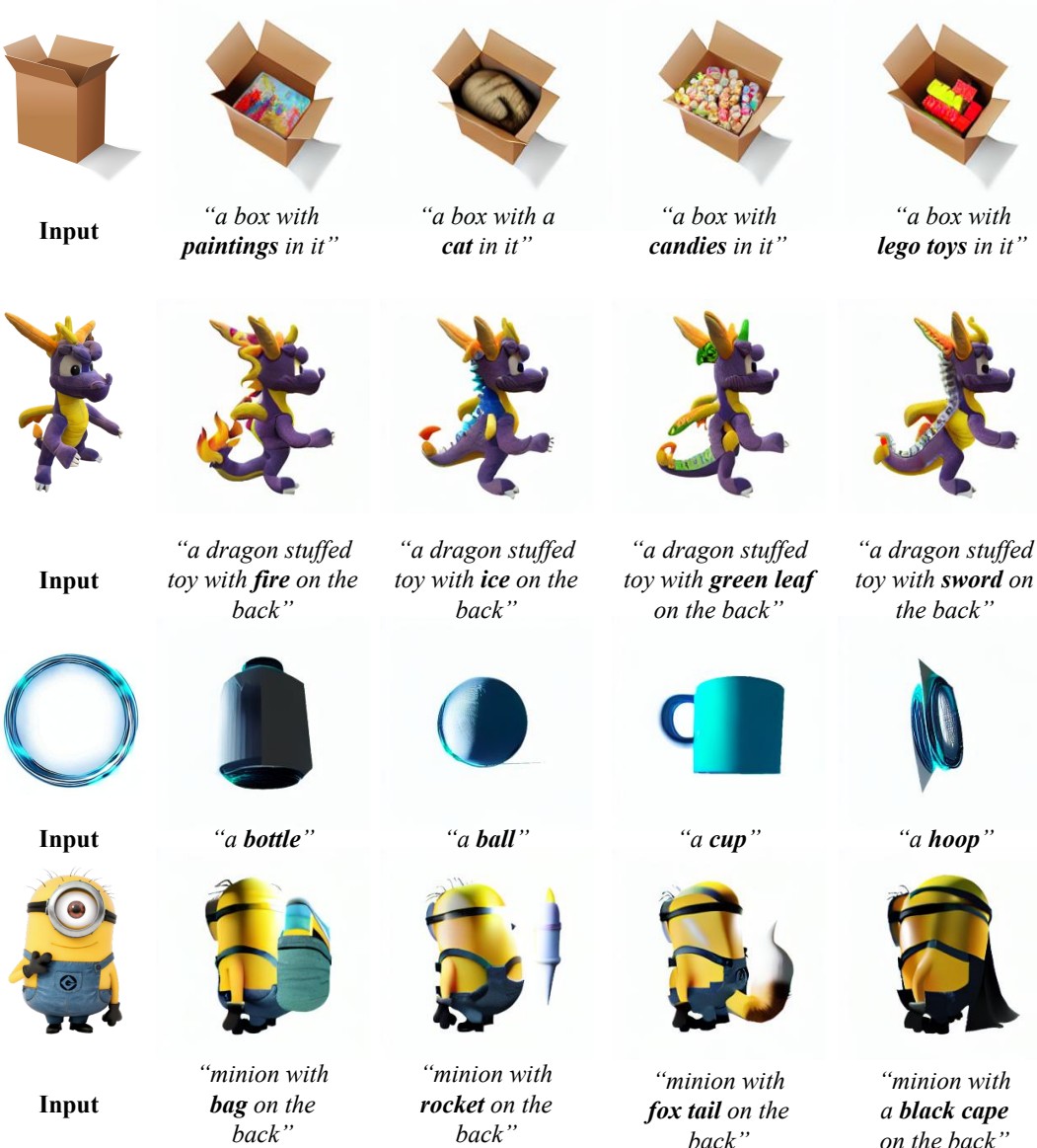

Figure 15: **More results demonstrating the increased NVS controllability with TOSS.**

# H QUALITATIVE COMPARISON ON MULTI-VIEW CONSISTENCY

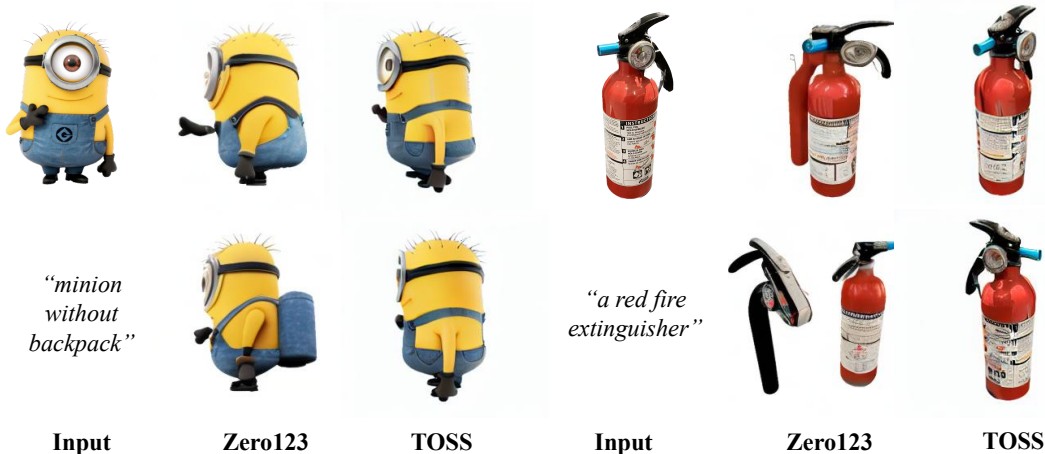

Figure 16: **Qualitative comparison on multi-view consistency**.

# I MORE NVS RESULTS

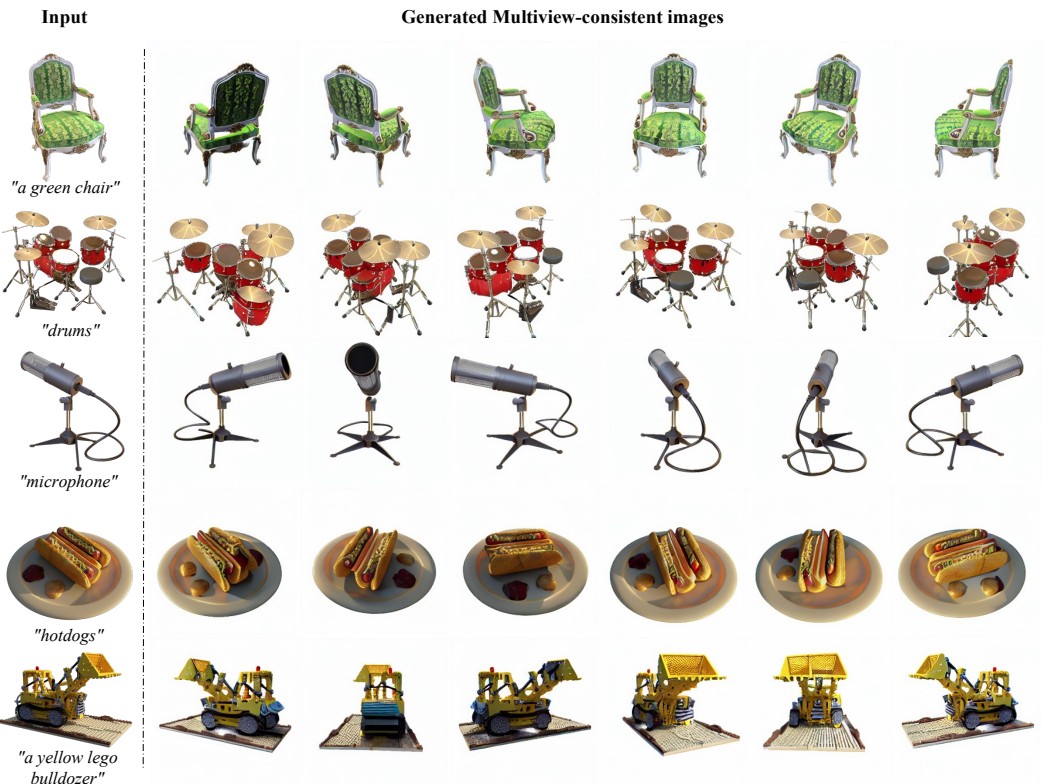

Figure 17: **NVS examples using TOSS on Synthetic NeRF dataset (Mildenhall et al., 2021)**

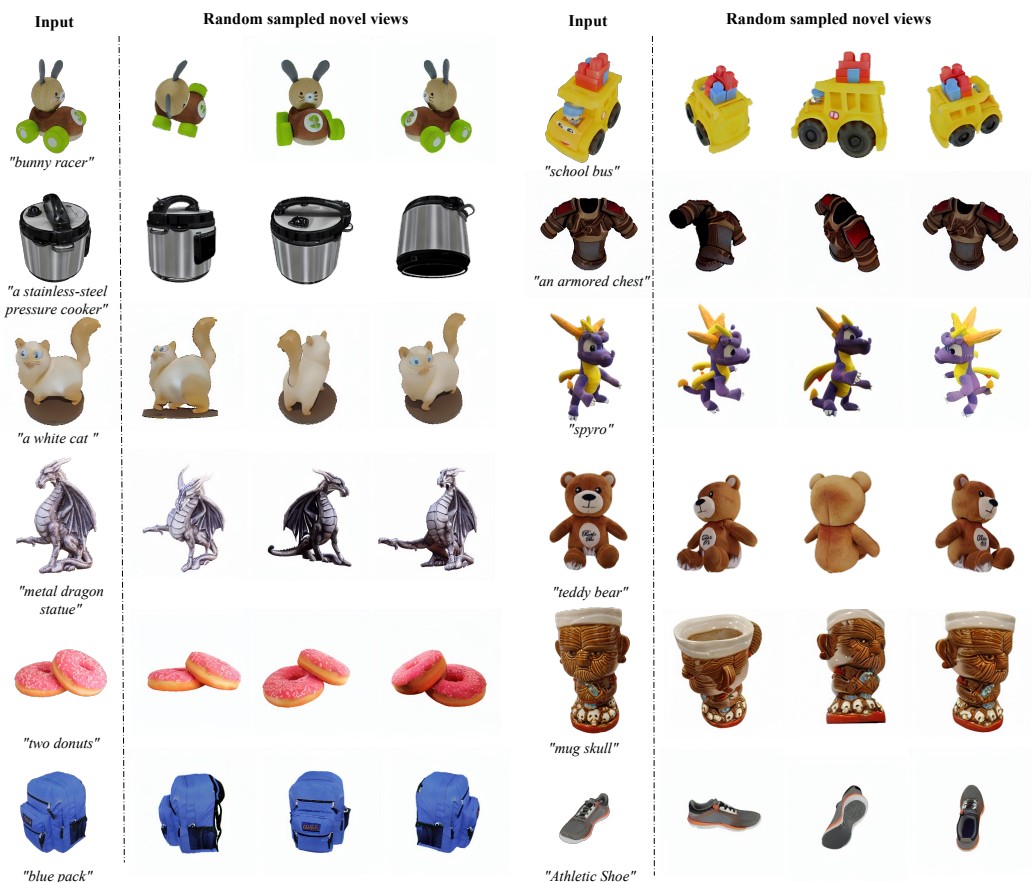

Figure 18: **Random sampled novel views using TOSS**. Input images are from the GSO dataset (Liu et al., 2023c) and Zero123 (Liu et al., 2023b) released on GitHub.

## J    MORE 3D CONSTRUCTION RESULTS

**Input**                                    **3D Reconstruction**

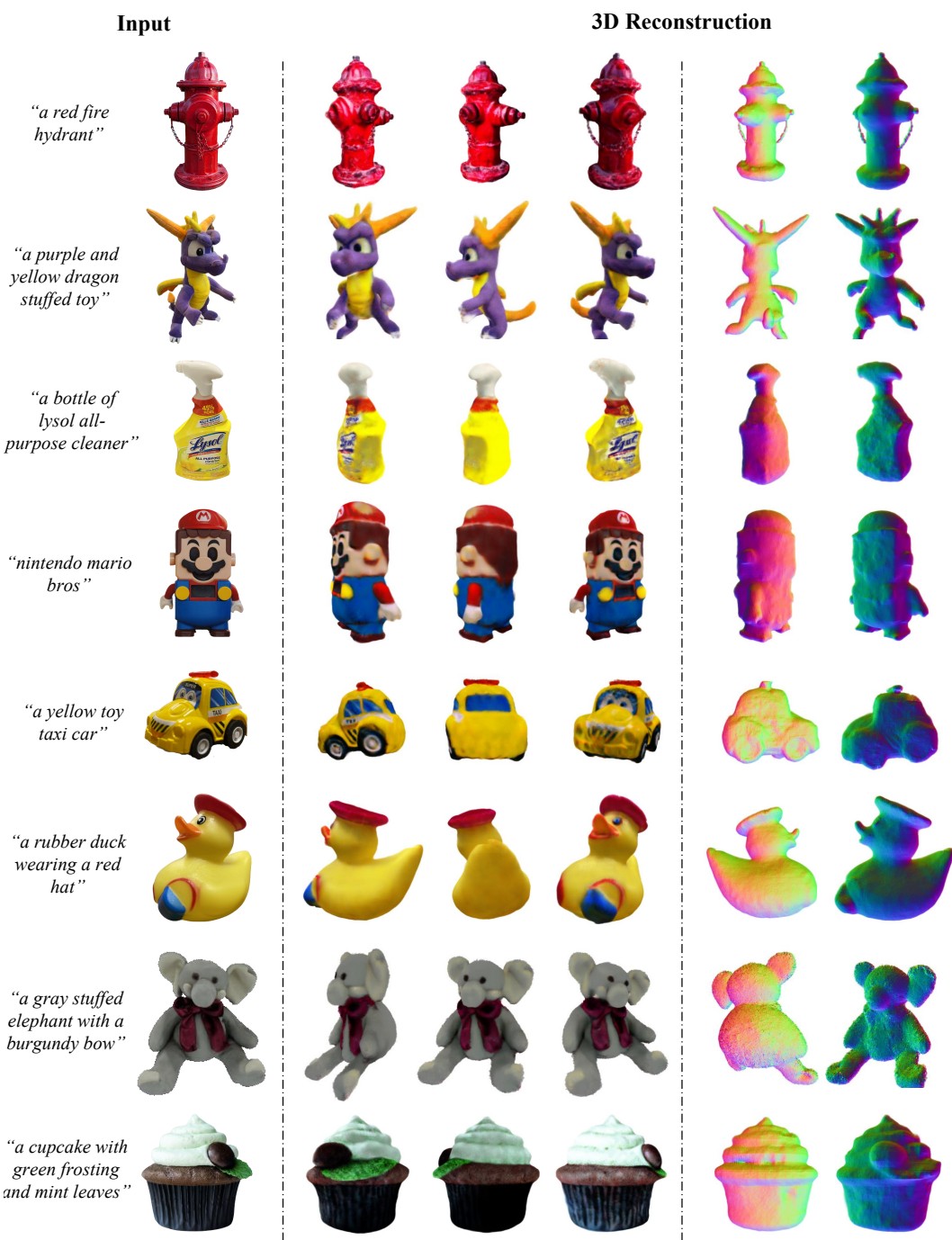

*"a red fire hydrant"*

*"a purple and yellow dragon stuffed toy"*

*"a bottle of lysol all-purpose cleaner"*

*"nintendo mario bros"*

*"a yellow toy taxi car"*

*"a rubber duck wearing a red hat"*

*"a gray stuffed elephant with a burgundy bow"*

*"a cupcake with green frosting and mint leaves"*

Figure 19: **3D reconstruction results on in-the-wild images**.

