# OpenReview forum: "TOSS: High-quality Text-guided Novel View Synthesis from a Single Image"
_ICLR.cc/2024/Conference — ICLR 2024 poster_

### Official Review · Reviewer_LBZY · 2023-10-23

**Soundness:** 3 good
**Presentation:** 2 fair
**Contribution:** 3 good
**Rating:** 6
**Confidence:** 4

**Summary:**

This paper proposes TOSS, which is an image-to-image diffusion model. Comparing to pioneering work like Zero123, TOSS is different because it's not only conditioned on input image and relative camera pose, but also an optional text prompt.

Since single image to other views are naturally an ill-pose problem, the authors claim that text prompt can provide more detailed guidance, which allows for increasing plausibility and controllability, and consequently better quality. In TOSS authors also analyze the drawbacks of previous conditioning mechanism and propose a dense attention mechanism for improved performance.

Authors provide a valid pipeline for automatic captioning and train the TOSS model on the large-scale Objaverse dataset. The extensive experiments prove the validness of each components and the superior performance of the proposed TOSS model.

**Strengths:**

In general, this paper is motivated, well-organized and provide valuable results. In detail I find paper's strength in several aspects:

1. The design of dense attention mechanism and the necessity of text as input are well motivated, with clear reason and proof.

2. The experiment part is sufficient. Extensive experiments have been done on not only for the comparison and main results, but also ablation on a wide range of components.

3. The authors also propose some valuable practices about image-to-image diffusion model, for example the expert denoiser design.

**Weaknesses:**

### 1. Over-claim multiview-consistency ###

I think the fundamental weakness for this paper is over-claim. In Sec.3.2.1, authors list two advantages of introducing text-prompt: plausibility and controllability, and I agree these two points. However, in the other parts of the paper, for example the Fig.1's caption, authors list *multiview-consistency* as another improvement, which I totally disagree.

Fundamentally, TOSS is like the Zero123 but with more input condition, it doesn't adopt some explicit design to guarantee the multiview-consistency like in SyncDreamer[1], so naturally it shouldn't be multi-view consistent. And although authors claim this in a lot of places in paper, I didn't find explanation towards this point.

Also, in Fig.13, while author titled it as "Generated Multiview-consistent images", it's actually not. For example in the provided microphone samples, I can clearly find it's not consistent across different views. So I think authors should revise the manuscript and avoid over-claim on being multivew-consistent.

### 2. Emphasize the controllability ###

Another minor weakness is I hope authors can emphasize more on the TOSS's controllability. I would like to see the authors provide more visualization results where from the same input image, different text can generate different novel views, currently the results are mainly in supplementary and I think it's better to emphasize this ability and move the part to the main paper.

[1] Liu, Yuan, et al. "SyncDreamer: Generating Multiview-consistent Images from a Single-view Image." arXiv preprint arXiv:2309.03453 (2023).

**Questions:**

In Sec4.5, what is the detail meaning of "replace the SD v1.4 model". Does it mean you replace the VAE encoder and the decoder? Does it replaced only in inference or in the training and re-train the model?

---

> ### Author Response · Authors · 2023-11-20
> **Response to R4**
>
> Thank you very much for the insightful feedbacks. We’ve provided more results according to your comments in the new revision.
> 1. **Over-claim multiview-consistency**: We agree that we do not introduce explicit modules to encourage multiview-consistency like in SyncDreamer. **However, we claim that text prompts improve the multiview-consistency with semantic constraints.** We support the conclusion with both qualitative and quantitative comparisons. **For qualitative comparison**, we refer to the minion in Figure 16 as an example. As we provide a front view to be the input condition image, Zero123 generates severe multi-view inconsistent results (minion with/without backpack), even under similar camera poses. However, our method significantly improves multi-view consistency with semantic constraint from text prompt "minion without backpack". **For quantitative comparison**, we evaluate 3D consistency scores in Table 2, which shows that our method indeed improves the multiview-consistency. In comparison to SyncDreamer, our improvement on multi-view consistency is not limited by azimuth or elevation angles. We also agree that it's extremely challenging to achieve 100% consistency with single-view novel view synthesis as eventually, this problem is severely under-constrained. We will revise the manuscript to avoid over-calim on being multivew-consistent but rather with improvements on multivew-consistency. We'll also cite more works on explicit multiview-consistency improvement for readers' reference.
> 2. **Emphasize the controllability**: Thanks for pointing this out. We now provide more results on controllability in Figure 15.
> 3. **Replace SD model for better performance**: We replace the initilazation weights of UNet and finetune it with higher resolution images. The other parameters such VAE encoder and decoder remain frozen.

---

> > ### Comment · Reviewer_LBZY · 2023-11-22
> >
> > I have read all the reviews and author's response. I think overall this is a good paper and the minor problems can be fixed in revision. Therefore I'll keep my rating the same.

---

### Official Review · Reviewer_Givb · 2023-10-27

**Soundness:** 3 good
**Presentation:** 3 good
**Contribution:** 3 good
**Rating:** 6
**Confidence:** 2

**Summary:**

This paper proposes a TOSS that is text to the task of novel view synthesis (NVS) from a single image. Compared to a previous method (Zero123), TOSS utilizes a text as high-level semantic information to constrain the NVS solution space. It is based on the text-to-image Stable Diffusion pre-trained on the large-scale dataset. The proposed method achieves plausible results and those are controllable. The effectiveness of the proposed method is validated on the dataset.

**Strengths:**

[Novelty]

- This paper utilizes diffusion prior and text embedding features with cross attention to perform NVS using a single image.

- This improves reconstruction performance for unseen areas and allows for consistent image generation.

[Quality]

- Overall, the paper is easy to follow and well written.

- This paper reports the impact of each module of the proposed method in terms of performance as appropriate.

[Clarity]

- The paper is clearly written.

- Motivation and explanation of each proposed module are reasonable.

[Significance]

- As a study that improves upon existing research, I think there are many elements that can be utilized in follow-up studies.

**Weaknesses:**

- Compared to just using cross attention (CLIP embeddings), it seems that it needs to analyze various perspectives such as computational cost or memory.

- Also, the analysis of the effectiveness of Expert Denoioser seems to be somewhat lacking.

- There are experiments that show the quantitative effectiveness of each of the proposed modules, but the qualitative, in-depth analysis is somewhat lacking.

- This paper is somewhat limited by the fact that comparative experiments were conducted with Zero123 only. I think it would be a better paper to compare NVS with a single image, or a diffusion-based model adapted to a given task, even if it is not exactly the same task.

- There is no section 3.2.2, so there is also no need for section 3.2.1. It is better to use just section 3.2.

**Questions:**

- In Table 1, there are no experiments for "inference w/o text" and no experiments for "w/ expert denoisers" for 160M?

---

> ### Author Response · Authors · 2023-11-20
> **Response to R3**
>
> Thank you very much for the insightful questions and suggestions. We’ve provided more results according to your comments in the new revision.
> 1. **Time/Memory cost**: As suggested, we analyze time/memory cost per module. (1) **Token-level attention** maintains the same time complexity but requires slightly more memory consumption for the VAE embeddings of the condition image. (2) **Text prompt attention** raises the attention token number from CLIP embedding's 1 to 77. In experiments, we evaluate time/memory cost on Nvidia A100 (80G) GPU with batchsize 128. During mixed-precision training, our method costs 2.13s and 52G per batch, while Zero123 costs 1.61s and 47G. **It's worth noting that our method is sufficiently stable, allowing us to employ half-precision training without any performance degradation, while Zero123 does not exhibit the same stability.** Therefore time/memory cost of out method could further decrease to 1.21s and 45G per batch with half-precision training, which is more efficient than Zero123.
> 2. **Effectiveness of expert denoisers**: For completeness, we follow the reviewer's suggestion and supplement the results of expert denoisers under the 160M setting in Table 1, showing quantitatively better results of expert denoises across different training budgets.
> 3. **More comparative experiments**: Thank you for the suggestion, we provide the results of DietNerf and Image Variation on GSO datasets for more comparative experiments in Table 1.
> 4. **Section 3.2.1**: Thanks a lot for pointing this out. We have removed the title of section 3.2.1 in the revision.
> 5. **"inference w/o text" and "w/ expert denoisers" experiments for 160M setting**: We've now included results under the 160M setting in Table 1.

---

> > ### Comment · Reviewer_Givb · 2023-11-22
> > **Thank you for the feedback.**
> >
> > Considering the opinions of other reviewers and author's, I keep my original positive score.

---

### Official Review · Reviewer_wyYi · 2023-10-28

**Soundness:** 3 good
**Presentation:** 3 good
**Contribution:** 2 fair
**Rating:** 6
**Confidence:** 4

**Summary:**

Novel view synthesis from a single RGB image is an under-constrained problem, and Zero123 solves this problem by training a view-conditioned latent diffusion model. However, Zero123 treats it as a pure image-to-image translation problem and thus suffers from pixel-level inconsistency problems. In this paper, TOSS adds text as high-level semantic information and, more importantly, proposes the cross-attention mechanism for better 3D consistency. Through exhaustive experiments, TOSS outperforms baseline Zero123 with more plausible and multiview-consistent NVS results, thus leading to substantial improvement in 3D reconstruction.

**Strengths:**

1. The proposed dense cross-attention mechanism is quite reasonable, and the explanation, especially in Figure 3, is persuasive. In the experiment section, both quantitative and qualitative results of novel view synthesis demonstrate the effectiveness of this module. More importantly, it significantly improves the 3D consistency, reflected in better 3D reconstruction results.
2. The additional text conditions increase the controllability of the TOSS model. Hence, text prompts lead to diverse NVS results as presented in the paper.
3. This paper is well-written and presented with a clear architecture. For the contributions summarized in the introduction, the corresponding experiment support can almost be found in the experiments.

**Weaknesses:**

1. The authors claim that the text prompts increase the plausibility of novel view synthesis results. However, only the corresponding quantitative results (especially in the ablation study) can be found in the paper. Apart from the illustration in Figure 1, more qualitative results should be presented to support this claim.
2. From the comparisons in Tables 1&2, TOSS only brings minor improvements over baseline method Zero123. Actually, prior work One-2-3-45 presents an interesting heatmap in Figure 4, showing the PSNR values are significantly affected by the relative azimuth and elevation angles. Hence, the authors should elaborate on how the results in Tables 1&2 were obtained to avoid unfair comparison.
3. For the 3D reconstruction, the authors should also compare with relevant methods such as Magic123 and Consistent123.

**Questions:**

1. During the cross-attention process, have you tried other orders or combinations of text, image, and camera pose?
2. Expect more detailed elaborations on expert denoisers in the paper, in particular the motivation.

---

> ### Author Response · Authors · 2023-11-20
> **Response to R2**
>
> Thank you very much for the insightful questions and suggestions. We’ve provided more results according to your comments in the new revision.
> 1. **Plausibility results**: In Figure 14 we've provided more qualitative results emphasizing plausibility improvements via text prompts.
> 2. **Camera pose details for fair comparison**: For all the NVS results, we **randomly sample** the camera pose similar to how Zero123 renders the Objaverse objects for training. That means **camera poses are not strictly limited in certain relative azimuth or elevation angles**, which is quite different from Sycdreamer[1]. Furthermore, we use the same sets of camera poses for all methods for fair comparison.
> [1]Liu, Yuan, et al. "SyncDreamer: Generating Multiview-consistent Images from a Single-view Image." arXiv preprint arXiv:2309.03453 (2023).
> 3. **More 3D reconstruction baselines**: We've further included the comparison to Magic123 in Table 3. Unfortunately, the code of consistent123 is not open-sourced yet, we plan to add its result and comparisons to more methods upon availability. As Magic123 utilizes Stable Diffusion and Zero123 separately without joint training on image and prompts, the quality of its results is highly reliant on the quality of text prompts. Therefore we use the same Blip2-generated captions across all methods for fair comparison.
> 6. **Ablation on cross attention process**: We supplement results of three ablation experimets on cross attention process below: (1) reverse the order of token-level attention and text prompt attention; (2) inject the camera pose information to text prompt attention instead of to token-level attention; (3) replace the token-level attention with gated attention and train it as an adapter with other parameters freezed. We can conclude from results of (1)(2) that different order of attention modules or different camera pose injecting choices lead to decrease in performance. Besides, experiment (3) reveals that **finetuning significantly enhances model performance compared to adapter training**.
>
>
>     | Method | PSNR | SSIM |
>     | -------- | -------- | -------- |
>     | (1) Reverse the order of  attention     | 16.63     | 0.8328     |
>     | (2) Inject the camera pose with text prompts     | 16.84     | 0.8275     |
>     | (3) Gated attention | 13.51 | 0.7105 |
>     | TOSS | 16.95 | 0.8369 |
>
> 5. **More detailed elaborations on expert denoisers**: We discover that during sampling, novel view pose is dedicated at large timesteps and details are further refined at small timesteps. The observation is similar to the observation in diffusion-based 2D image generation that structure forms during early sampling steps and later sampling steps are responsible for image details. Therefore, we draw inspiration from ensemble of expert denoisers [1] to train two expert denoisers that are specialized for denoising at different timestep intervals.\
> [1] Balaji, Yogesh, et al. “ediffi: Text-to-image diffusion models with an ensemble of expert denoisers”.

---

> > ### Comment · Reviewer_wyYi · 2023-11-23
> > **Response to the authors**
> >
> > I have read the authors' response, which settles most of my concerns. I suggest to include these additional results in the revision.  Considering the opinions of other reviewers, I decide to keep my original score and am leaning to accept this paper.

---

### Official Review · Reviewer_gFkU · 2023-11-02

**Soundness:** 3 good
**Presentation:** 3 good
**Contribution:** 3 good
**Rating:** 6
**Confidence:** 5

**Summary:**

This paper proposes to use text as high-level semantic information to constrain the NVS solution space. With this text, the proposed method can generate the multi-view consistency images. Meanwhile, it proposes dense cross-attention to align the reference and target image.

**Strengths:**

1. The whole idea is Intuitive and effective.  Using the text as semantic guidance to preserve multi-view consistency is reasonable since the ref image can not provide enough information. Meanwhile, since the proposed model uses the text prompt, it can finetune the SD directly, which preserves the ability of generation.
2. The attention strategy is reasonable, which will help these two images align better.

**Weaknesses:**

1. My main concern is Whether the texts bring enough improvement.  As shown in Table 4, I find "Token-level attention"  improves the model the most, while the "Text prompt" is not the best.  Meanwhile, the author should show these cases visually: 1. Zero123 + Token-level attention 2. Zero123 + Text prompt.
2. I'm curious about the effect of this model on the human portrait since the Zero123 performance is bad in human portraits (Dataset limitations). The author can show two cases visually: 1. Taylor Swift (you can select an image of Taylor Swift as input) 2. Young male (you can randomly select a young male image).

If the authors can address my question properly, I will be glad to improve my grade.

**Questions:**

see the weakness

---

> ### Author Response · Authors · 2023-11-20
> **Response to R1**
>
> Thank you very much for the insightful questions. We’ve provided more results according to your comments in the new revision.
> 1. **Contribution of text prompt**: As requested, we've provided additional visual results showing contributions of the introduced modules in Figure 13. We can see that even though "Token-level attention" indeed improves the model the most in terms of metrics, the "Text prompt" **still significantly enhances the quality of the generated novel views** visually, including more plausible and intricate details (for example the lights on the top of the police car). We would like to point out that **the contribution of text prompts exceeds what the metrics demonstrate**. The reviewer's intriguing question precisely aligns with the observation in 3DiM[1]: **The standard metrics (PSNR/SSIM) are not sufficient to effectively evaluate geometry-free models for view synthesis**. Since NVS from a single image is an ill-posed problem (e.g. novel view generation of the backview given the front view), sometimes a plausible generation  which is inconsistent with the ground truth may result in a decrease in PSNR/SSIM. \
> [1]Watson, Daniel, et al. Novel View Synthesis with Diffusion Models. Oct. 2022.
> 2. **Effect on Human Portrait**: Thank you very much for the intriguing question. We provide additional 3D reconstruction results on Taylor Swift, young male and more in Figure 20. We acknowledge that there remains a quality gap between our method and the human-centric approaches, but **our results are of higher quality than zero123**. We agree with the reviewer that the quality concern in human portraits comes from the pretrain datasets (Objaverse), which lack sufficient human data. Therefore we believe the issue can be alleviated by finetuning on human-centric datasets which remains for future work.

---

> > ### Comment · Reviewer_gFkU · 2023-11-22
> > **Reponse to Authors**
> >
> > I think the authors address my first concern well, but it does not show good performance in human portraits due to the dataset limitation. So, I keep my original rating, I think this work can be accepted, but not to a high degree(i.e., oral or spotlight).

---

### Author Response · Authors · 2023-11-20
**Overall Reply**

We would like to thank all the reviewers for their constructive comments and valuable feedbacks. We are encouraged by the recognition that "the whole idea is intuitive and effective", "the proposed dense cross-attention mechanism is quite reasonable and persuasive", "the paper is well-written and presented in a clear architecture", "a study that can be utilized in follow-up studies", "the paper is motivated, well-organized with valuable results".
We individually respond to each reviewer’s comments below. Discussions and results will be added to new revisions.

---

### Meta-Review · Area_Chair_sgRN · 2023-12-04

**Metareview:**

Summary: The paper proposes to explicitly use text to narrow down the solution space of novel view synthesis (NVS). In contrast to prior works which handle NVS as an image-to-image translation, this paper utilizes a well motivated cross-attention mechanism to maintain 3D consistency. Overall, the method is motivated, well organized, and clearly written. The effectiveness of the methodology is validated through extensive experiments, as unanimously acknowledged by reviewers.
Weakness: A few missing comparisons and ablations as raised by the reviewers, while some have been addressed during the rebuttal.

**Justification For Why Not Higher Score:**

The paper has a few missing ablations and comparisons.

**Justification For Why Not Lower Score:**

The paper tackles an interesting problem space – multi-view synthesis from a single RGB image. It makes sufficient innovations that have been validated through extensive experiments. It also received 4x above acceptance threshold.

---

### Decision · Program_Chairs · 2024-01-16

Accept (poster)